# Regulating translation in aging: from global to gene-specific mechanisms

Mathilde Solyga [iD][1], Amitabha Majumdar [iD][2] & Florence Besse [iD][1✉]

## Abstract

**Aging is characterized by a decline in various biological functions that is associated with changes in gene expression programs. Recent transcriptome-wide integrative studies in diverse organisms and tissues have revealed a gradual uncoupling between RNA and protein levels with aging, which highlights the importance of post-transcriptional regulatory processes. Here, we provide an overview of multi-omics analyses that show the progressive uncorrelation of transcriptomes and proteomes during the course of healthy aging. We then describe the molecular changes leading to global down-regulation of protein synthesis with age and review recent work dissecting the mechanisms involved in gene-specific translational regulation in complementary model organisms. These mechanisms include the recognition of regulated mRNAs by *trans*-acting factors such as miRNA and RNA-binding proteins, the condensation of mRNAs into repressive cytoplasmic RNP granules, and the pausing of ribosomes at specific residues. Lastly, we mention future challenges of this emerging field, possible buffering functions as well as potential links with disease.**

**Keywords** Aging; Post-transcriptional Regulation; Tanslation; RNA; RNA-Binding Proteins
**Subject Categories** Translation & Protein Quality; RNA Biology

## Introduction

Aging is associated with physiological changes that affect most biological functions and increase susceptibility to diseases. Distinguishing between "driver and passenger mechanisms of aging" (de Magalhaes, 2024) is a difficult task, but a number of cellular and molecular processes that are functionally contributing to the aging process have been defined (Lopez-Otin et al, 2023). Among the primary hallmarks of aging are epigenetic alterations that affect nuclear DNA methylation patterns, post-translational modifications of histones and chromatin remodeling and thus profoundly modify gene expression by altering transcription programs (Sen et al, 2016). While regulation of nuclear transcription is undoubtedly a critical step in the control of gene expression,

other post-transcriptional mechanisms affecting pre-mRNA fate and outputs through control of isoform selection (Ule and Blencowe, 2019), RNA stability (Houseley and Tollervey, 2009), RNA localization (Das et al, 2021) or RNA translation (Kong and Lasko, 2012) have been described (Medioni and Besse, 2018). These regulatory steps are mediated by specialized cellular machineries (spliceosomes, ribosomes, exosomes, transport machineries and so on) recruited in *trans* to RNA molecules, but also involve *cis*-regulatory marks found in RNA molecules that modulate the recruitment and/or the efficiency of these machineries (Fig. 1). These marks can be RNA modifications (e.g., m6A methylation, RNA editing) (Roundtree et al, 2017), regulatory sequences located in the non-translated 5′ or 3′UTR region of mRNAs (e.g., binding sites for RNA-binding proteins or complementary noncoding RNAs) (Mayr, 2019) or specific coding sequence features (e.g., codon composition) (Hanson and Coller, 2018).

Despite expanding knowledge on the mechanisms controlling RNA fate, the role and regulation of post-transcriptional processes in the context of aging have so far been poorly investigated. However, recent large-scale integrative studies combining transcriptome, proteome and translatome profiling have revealed that aging is characterized by a significant loss of correlation between RNA and protein levels, indicating the importance of post-transcriptional regulatory processes in shaping the proteomes of aging organisms (Cellerino and Ori, 2017). Functional studies performed in a number of complementary model organisms (Box 1) have also underscored the contribution of these processes to age-related physiological changes. In this review, we describe recent work about the uncoupling between RNA and protein expression during aging and focus on the mechanisms contributing to age-dependent changes in RNA translation patterns. These mechanisms include both general modulation of the protein synthesis machinery and more specific processes that target selected subsets of transcripts.

## Aging is characterized by a progressive uncoupling of RNA and protein levels

Recent multi-omics studies combining RNA-seq and Mass-Spectrometry on a range of model organisms (human, mouse, killifish, nematode, and so on) and tissues such as kidneys, heart, brain, etc, have uncovered that age-dependent changes in protein levels overall poorly correlate with changes in corresponding RNA levels (Di Fraia et al, 2024;

[1]Université Côte d'Azur, CNRS, Inserm, Institut de Biologie Valrose, Nice, France. [2]National Centre for Cell Science, Savitribai Phule Pune University Campus, Pune, Maharashtra, India. ✉E-mail: florence.besse@univ-cotedazur.fr

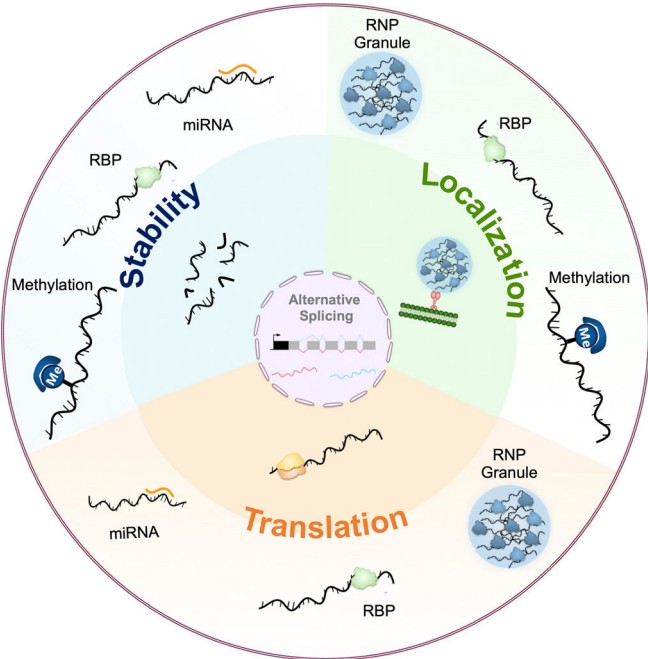

**Figure 1. Overview of post-transcriptional regulatory mechanisms.**

Once exported from the nucleus to the cytoplasm, mRNAs undergo post-transcriptional regulation at three different levels: stability, localization, and translation. These processes, represented in the inner circle, are modulated by *trans*-acting factors such as RNA-Binding Proteins (RBPs) or miRNAs, but also by epigenetic processes such as methylation or condensation into so-called ribonucleoproteic (RNP) granules (outer circle). RBPs regulate mRNA fate at different levels: translation, localization and stability. RBPs can have an opposite impact on translation by either promoting the recruitment of the translation machinery or preventing it through competition with translation initiation factors. Similarly, RBPs play a dual role on mRNA stability: they can protect mRNAs through capping and polyadenylation modulation or induce their cleavage and decay. Lastly, RBPs can recruit molecular motors to promote mRNA transport and remote translation. miRNAs hybridize to target mRNAs via complementary base pairing, leading to the recruitment of the RNA-induced silencing complex (RISC). This interaction can result in either translational repression or mRNA degradation. Methylation modifies the structure of RNA molecules and their binding affinity for RBPs, which can impact their stability or localization. RNP granules mediate the clustering of mRNA subsets into RBP-rich condensates that are depleted of the translation machinery and can be transported over long distances.

**Box 1    Common model organisms in aging studies**

The **yeast** Saccharomyces cerevisae is a unicellular eukaryotic organism whose genome size is ~12 Mb, with around 6000 protein-coding genes. In this model, aging is mostly defined as replicative aging and measured by the number of times a mother cell has divided. It can also be defined as chronological aging and measured by how long cells stays alive under nutrient deprivation (typically 1 week).

The **nematode** Caenorhabditis elegans has a genome size of ~97 Mb, with around 19,800 protein-coding genes. In this invertebrate model, age is defined as the number of days post-eclosion and the average lifespan is 2–3 weeks.

The **fly** Drosophila melanogaster has a genome size of ~180 Mb, with around 13,600 protein-coding genes. In this invertebrate model, age is defined as the number of days post-hatching and the average lifespan is 2–3 months.

S. cerevisiae, C. elegans, and Drosophila all are genetically tractable model systems in which gene function can easily be manipulated via diverse tools enabling tissue-specific gene overexpression or inactivation. They are also amenable to systematic genetics and drug screenings. Providing evolutionary support to the molecular mechanisms underlying aging, major genetic pathways involved in vertebrate aging, such as the Insulin or the mTor pathways, have been initially discovered and dissected in these models.

The turquoise **killifish** Nothobranchius furzeri has a genome size of ~1.2 Gb, with around 28,500 protein-coding genes. This vertebrate model attains sexual maturity within 3–4 weeks of hatching and has an extremely short lifespan of 4–6 months. It genetically tractable, although its generation time is 40 days.

The mice Mus musculus has a genome size of ~2.5 Gb, with around 30,000 protein-coding genes, 85% of them having orthologues in the human genome. This mammalian model attains sexual maturity within 4–7 weeks and lives for 2.5–3 years. This model is amenable to gene manipulations and editing. It possesses a developed tool kit for generating transgenics, knockins and knockouts, although generation time is 12 weeks. This model is used for screening of anti-aging drugs.

Beyond differences in lifespan, amenability to genetic manipulation, and transcriptome sizes, other parameters should be considered when studying post-transcriptional regulation in model organisms. Unicellular and invertebrate model organisms are particularly adapted to bulk transcriptome-wide studies where each datapoint represents an average population value, thus minimizing age-related inter-individual variabilities. Tissue-specific profiling cannot be done in unicellulars, and is easier to perform in vertebrate model organisms, where individual organs can easily be dissected and provide enough material for transcriptomics or proteomics.

Gerdes Gyuricza et al, 2022; Keele et al, 2023; Khatir et al, 2023; Ori et al, 2015; Takemon et al, 2021; Waldera-Lupa et al, 2014; Walther et al, 2015; Wei et al, 2015; Winsky-Sommerer et al, 2023). As demonstrated by comparing multiple time points, uncoupling of protein and RNA levels is much higher during aging than during development (Wei et al, 2015) and occurs progressively in the course of aging (Janssens et al, 2015; Kelmer Sacramento et al, 2020). It manifests as both positive uncoupling (increase in protein abundance relative to transcript level) and negative uncoupling (decrease in protein abundance relative to transcript level), the latter explaining up to 30–40% of the decoupling variance (Di Fraia et al, 2024; Wei et al, 2015).

Remarkably, combination of hierarchical clustering and definition of concordant and discordant classes of genes after analysis of RNA and protein expression levels in primate prefrontal cortex samples indicated that discordant gene classes tend to be enriched in regulatory and signaling functions. In addition, a strong conservation of mRNA and protein co-expression profiles was observed by comparing human and rhesus macaque samples, further suggesting functional importance (Wei et al, 2015). At the gene-specific level, uncoupling coefficients appeared to be highly heterogenous, leading to a decreased overall coordination between functionally related mRNAs. In the case of multi-molecular complexes, this translates into age-related alterations in the relative stoichiometry of constituent subunits (Gerdes Gyuricza et al, 2022; Janssens et al, 2015; Kelmer Sacramento et al, 2020). Together, a comparison of RNA and protein levels has highlighted the importance of post-transcriptional regulatory mechanisms in age-dependent alterations in gene expression.

Dysregulated proteostasis is a well-established hallmark of aging and could explain in part the observed RNA-protein uncoupling (Hipp et al, 2019; Kaushik and Cuervo, 2015; Lopez-Otin et al, 2023). The contribution of additional upstream mechanisms regulating translation efficiency has however emerged through

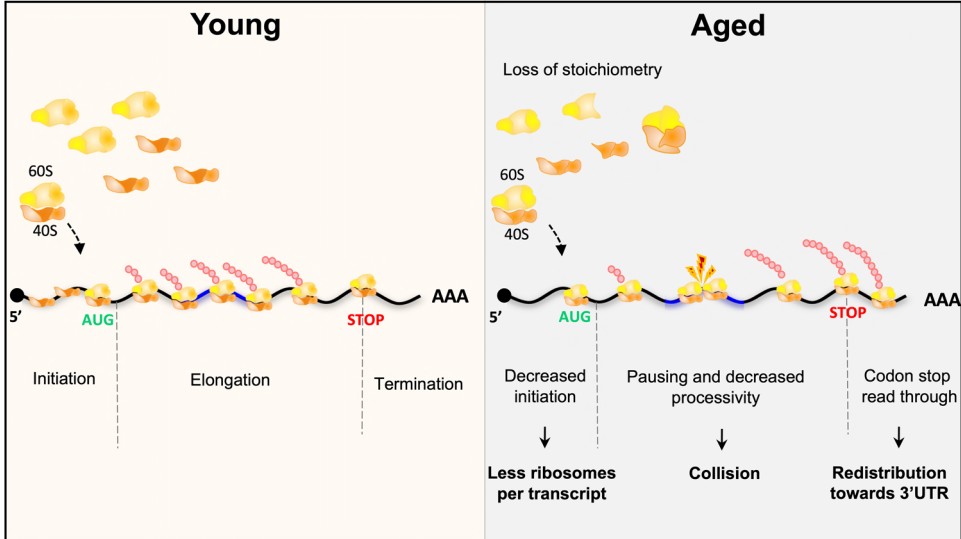

**Figure 2. Age-dependent alterations in the efficiency of the translation machinery.**

Schematic representation of the three main steps of elongation (left panel) and their alteration in aged individuals (right panel). During the initiation phase, the pre-initiation complex containing the small ribosomal subunit 40S scans the mRNA 5′ untranslated region (UTR) to find the start codon AUG. At this position, the large ribosomal subunit 60S is recruited to form a competent 80S ribosome. The loss of ribosomal protein stoichiometry observed with aging leads to a decreased initiation rate and a reduced number of 80S ribosome per transcript. During the elongation step, the polypeptide chain is synthesized through ribosome-mediated incorporation of amino acids via aminoacyl-tRNA codon recognition. In aged individuals, the rate of elongation, which reflects the processivity of the ribosome along the coding sequence, is decreased, thereby increasing the frequency of collisions. At the end of the coding sequence, the stop codon triggers the termination of translation and the dissociation of the ribosome from the mRNA. An increased frequency of STOP codon readthrough is observed in aging cells, inducing a partial relocalization of ribosomes to the 3′ UTR.

two complementary transcriptome-wide assays providing comprehensive snapshots of tissue translatomes: polysome profiling, in which polysome-associated mRNAs are isolated through sucrose density fractionation before RNA-sequencing; and ribosome-sequencing (Ribo-seq), in which ribosome-associated mRNA fragments (or ribosome footprints) are specifically purified and sequenced (Brar and Weissman, 2015; Iwasaki and Ingolia, 2017). Comparison of Ribo-seq and proteomic data from young and aged rat brain and liver, for example, revealed the existence of hundreds of transcripts exhibiting significant changes in translation efficiency during aging and an overall good correlation between changes in protein levels and translation efficiency (Ori et al, 2015). An independent study performed in killifish brain also observed that variations in translational efficiency are better suited to explain changes in protein abundance than variations in RNA levels (Di Fraia et al, 2024), further suggesting the importance of translational regulation in aging. Of note, strong tissue-specific differences were seen when comparing gene expression profiles, both in terms of the identity of genes undergoing age-dependent changes and in terms of the extent of protein synthesis-dependent control, as up to 15% of the transcripts analyzed exhibited alteration in translation efficiency in the aging rat brain compared to only 2% in the liver (Ori et al, 2015). As further described below, it has become clear that age-dependent changes in translation efficiency result both from a general modification of the activity of the protein synthesis machinery and from translational regulation of selected subsets of transcripts (Fig. 2).

# Global decline in protein synthesis with aging

Pioneer experiments performed in the 1970s and 80s via metabolic incorporation of radiolabeled amino acids in a wide range of cell-free systems and tissues from vertebrates (rat, mouse) or invertebrates (*Drosophila*) have shown a significant decrease in the rate of global protein synthesis with aging (Blazejowski and Webster, 1983; Dwyer et al, 1980; Fando et al, 1980; Kim and Pickering, 2023; Layman et al, 1976; Ward and Richardson, 1991; Webster and Webster, 1979). More recent ribosome profiling experiments, in which samples from replicatively aged yeast, old *C. elegans* or rat brains were sedimented on a sucrose gradient to separate free RNAs from ribosome-engaged RNAs, have confirmed these initial discoveries and demonstrated a marked decrease in the global amount of RNA associated with ribosomes in old compared to young cells (Fando et al, 1980; Hu et al, 2018; Motizuki and Tsurugi, 1992; Stein et al, 2022).

The number of ribosomes loaded per mRNA molecule is controlled by regulation of two main regulatory steps of translation (Hershey et al, 2012; Kim and Pickering, 2023; Yuan et al, 2024): (i) the rate-limiting initiation step, in which the small 40S ribosomal subunit is recruited to the mRNA and then engages into 5′UTR scanning until reaching an initiation codon whereby the large 60S subunit is recruited to assemble the translation-competent 80S initiation complex; and (ii) the elongation step in which codon-specific aminoacyl-tRNAs are sequentially recruited

to the ribosome and their associated amino acids are covalently linked to the growing nascent peptide chain before translocation to the next codon. Thus, regulation of initiation and elongation rates can shift the balance between ribosome-free or monosome-associated RNAs (not or poorly translated RNAs) and polysome-associated RNAs (efficiently translated RNAs). Studies performed in different models have shown that both translation initiation and translation elongation are affected by aging (Anisimova et al, 2018; Kim and Pickering, 2023).

## Decrease in translation initiation

Historical experiments performed in cell-free extracts from young and old mouse livers have pointed to a decreased activity of the 40S ribosomal subunit in forming the initiation complex with aging (Nakazawa et al, 1984). More recent estimation of the amount of monosome-engaged versus polysome-engaged RNAs, combined with global profiling of ribosome occupancy along transcripts, further demonstrated a strong reduction of translation initiation in vivo in aged animals, whether replicatively aged yeasts or >10-day-old C. elegans (Hu et al, 2018; Stein et al, 2022).

### Decrease in the pool of functional ribosome subunits upon aging
Such a general decrease in translation initiation can be explained to a great extent by a loss of functional ribosomes, as the biogenesis of the protein synthesis machinery is altered in different ways during aging. First, levels of ribosomal proteins were found to be strongly downregulated in aging tissues of vertebrate and invertebrate models (Keele et al, 2023; Khatir et al, 2023; Ubaida-Mohien et al, 2019; Walther et al, 2015), a phenomenon that was identified as upstream and causal in a high-resolution systems-level model of aging, modeling the interdependence of gene expression changes occurring throughout replicative aging in yeast (Janssens et al, 2015). Second, decrease in protein levels is seen for many, but not all ribosomal proteins, leading to a progressive imbalance in the relative levels of the different ribosomal machinery constituents from yeast to mammals (Gerdes Gyuricza et al, 2022; Janssens et al, 2015; Keele et al, 2023; Kelmer Sacramento et al, 2020). Such a loss of ribosomal protein stoichiometry is associated with defects in ribosome assembly, as seen in size-exclusion chromatography experiments in which components of the ribosome co-eluted at lower-than-expected molecular weight in old killifish brain lysates (Kelmer Sacramento et al, 2020). Third, age-induced ribosomal protein imbalance creates a pool of orphan subunits at risk of aggregation. Consistent with this, various studies have shown that ribosomal proteins tend to accumulate in the insoluble fractions collected from different tissues and different organisms (Chen et al, 2024; David et al, 2010; Di Fraia et al, 2024; Harel et al, 2024; Kelmer Sacramento et al, 2020; Reis-Rodrigues et al, 2012), with a particularly high propensity to aggregate in organs such as the gut, the liver and the brain (Chen et al, 2024). Notably, the decrease in the pool of functional ribosomes may not only be explained by defective ribosome assembly, but also by defective ribosome recycling. Accumulation of isolated 3'UTR fragments loaded with ribosomes was indeed described in the aging rat brain in response to oxidative stress and altered translation termination (Sudmant et al, 2018). Furthermore, an increased frequency of stop codon readthrough was observed in aged Drosophila (Chen et al, 2020; Martinez-Miguel et al, 2021).

### Decreased amount and/or activity of translation initiation factors
The decrease in translation efficiency observed during aging has been explained in parallel by changes in the activity and levels of translation initiation factors. For example, the amount of eIF2, a factor essential for recruiting the initiator Met-tRNA$_i$ to the 40S subunit and further assembly of the 43S pre-initiation complex, was shown to decline with age and to linearly scale with protein synthesis in rat liver, kidney, lung and brain (Kimball et al, 1992). Age-dependent changes in the phosphorylation patterns of distinct translation initiation factors were also observed in rat brains (Ori et al, 2015) and yeast (Hu et al, 2018). Phosphorylation of eIF2α, a process known to prevent regeneration of active eIF2 and thus reduce protein synthesis (Rowlands et al, 1988; Sonenberg and Hinnebusch, 2009), was in particular shown to be induced during yeast replicative aging in response to the activity of the Gcn2 kinase (Hu et al, 2018). eIF2α phosphorylation has in fact been observed in various biological aging models, including C. elegans (Derisbourg et al, 2021b) and mouse liver and kidney (Ladiges et al, 2000). It represents the main target of the Integrated Stress Response (ISR) pathway, whose age-related increase in activity was proposed to fine-tune translation in the context of aging (Derisbourg et al, 2021a).

eIF4E, a factor recognizing the 7-methyl-guanosine 5′ cap structure of mRNAs and required for assembly of the eIF4F complex and further recruitment of the 43S pre-initiation complex, is also under tight control (Hershey et al, 2012; Sonenberg and Hinnebusch, 2009). Work performed in C. elegans, for example, has shown that not only the levels, but also the availability of IFE-2, the worm somatic eIF4E, decrease with aging (Rieckher et al, 2018). Specifically, the observed accumulation of IFE-2 into P-bodies, cytoplasmic condensates concentrating RNA molecules and RNA-binding proteins involved in RNA decay and translational repression (Ripin and Parker, 2023), was proposed to sequester IFE-2, thus contributing to the general decrease in protein synthesis with aging (Rieckher et al, 2018).

## Changes in translation elongation

Decreased efficiency of translation elongation has also been implicated in the overall reduction of protein synthesis during aging (Connors et al, 2008; Kim and Pickering, 2023). As revealed by comparing ribosome half transit time in cell extracts from young and aged rats, a significant decrease in the rate of peptide chain elongation was indeed observed in liver cells (Coniglio et al, 1979). Pulse-chase experiments performed in mice to estimate the in vivo rate of amino acid incorporation further confirmed this finding (Gerashchenko et al, 2021). In this work, the sequential application of harringtonine, an inhibitor that specifically blocks translation initiation and cycloheximide, an inhibitor that blocks translation elongation, was followed by Ribo-seq and reconstitution of ribosome coverage tracks, which revealed a near 20% decrease in the rate of elongation in the liver. Age-dependent decrease in the efficiency of peptide chain elongation was additionally observed in cell-free extracts prepared from Drosophila whole bodies (Webster and Webster, 1979), and attributed to a large extent to alteration in the binding of aminoacyl-tRNA to ribosomes (Webster and Webster, 1982).

As recruitment of aminoacyl-tRNAs to the translating ribosomes is mediated by the eukaryotic elongation factor 1 (eEF-1),

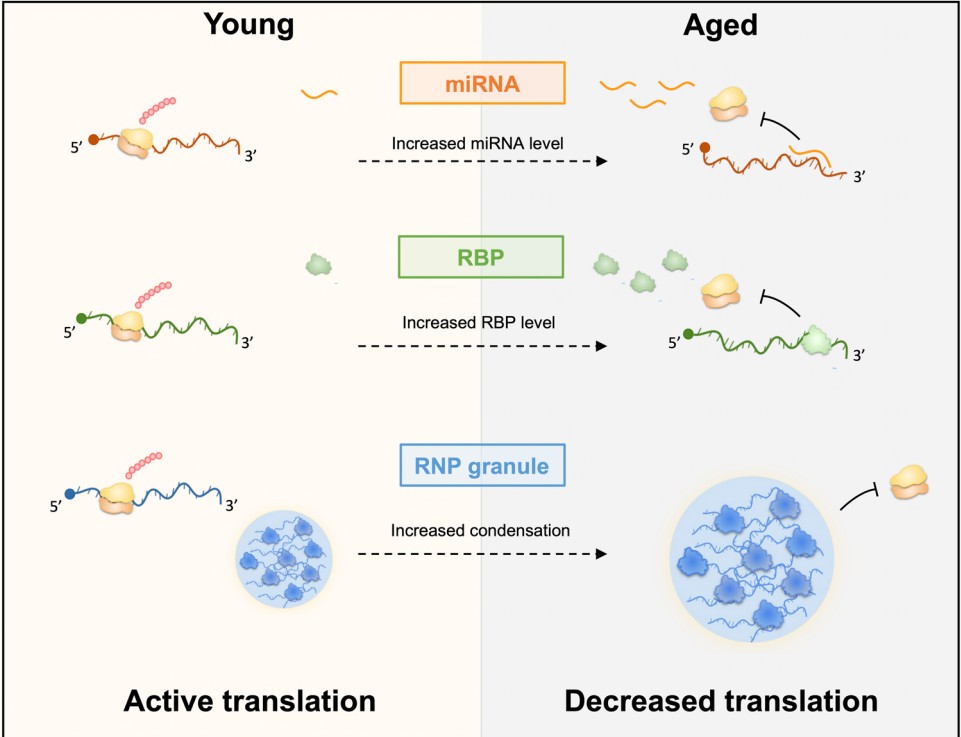

**Figure 3.** *Trans*-acting factors contributing to translational repression with aging.

Age-dependent increase in the level of *trans*-acting factors such as miRNA or RBPs (upper and middle parts, respectively) were shown to induce the translational repression of their target mRNAs. Condensation of RBPs and their associated mRNAs into bigger granules devoid of ribosomes (lower part) can induce their translational repression.

age-dependent changes in the levels and activity of eEF-1 were analyzed. In *Drosophila* cell extracts, a marked decrease in the synthesis of eEF-1 was apparent as early as day 7, that is, prior to the main drop in cellular protein synthesis (Webster and Webster, 1983). In rat, eEF-1 activity was found to be 30–40% higher in young compared to old brain and liver (Moldave et al, 1979). In addition, the activity of eEF-2, another elongation factor involved in the translocation of the peptidyl-tRNA from the A- to the P-site of the ribosome, was found to undergo age-related changes in rat liver cell-free extracts. Specifically, the activity of the eEF-2 kinase and the phosphorylation level of its target eEF-2, both associated with a decreased activity of eEF-2 and a reduction in peptide elongation, were shown to be more than 2-fold higher in extracts from aged rats compared to young ones (Riis et al, 1993). Together, these changes participate in the general reduction in the efficiency of the protein synthesis machinery with age.

## Mechanisms regulating the translation of specific subsets of transcripts with aging

In addition to the global decrease in translation efficiency observed with aging, specific signatures of mRNA-protein uncoupling were identified in transcriptome-wide sequencing studies (Di Fraia et al, 2024; Ori et al, 2015; Wei et al, 2015), pointing to gene-specific post-transcriptional regulatory processes. Although the mechanisms underlying such specific changes largely remain to be

explored, emerging studies have highlighted the role of distinct molecular pathways, the regulation of which impacts on the expression of classes of downstream transcripts in the context of aging. For example, longitudinal Ribo-seq analyses performed on livers of up to 32-month-old mice indicated that mRNAs containing 5′-terminal oligopyrimidine (5′-TOP) regulatory sequences responsive to changes in the activity of the mTor pathway (Thoreen et al, 2012) exhibit particularly strong translational repression in aging (Anisimova et al, 2020). As described below, several other mechanisms involving noncoding RNAs (3.1), RNA-binding proteins (RBPs) and their condensation (3.2 and 3.3), or coding sequence properties (3.4) have been dissected in further detail (Fig. 3 and Table 1).

### Age-dependent changes in levels of miRNAs

miRNAs are small noncoding RNAs of 19–24 nucleotides in length that recognize complementary sequences in the 3′UTR regions of cellular mRNAs and repress their expression by inducing their degradation or translational repression (Shang et al, 2023). Each miRNA can simultaneously regulate multiple targets, thereby potentially co-targeting different components of functional networks involved in specific biological processes. A number of transcriptomics studies, performed over the years in complementary model organisms and tissues, have indicated that dozens of miRNAs are differentially expressed (either up- or downregulated) during aging (Ibanez-Ventoso et al, 2006; Inukai et al, 2012; Kato et al, 2011; Liu et al, 2012; Somel

**Table 1. Table listing the cited *trans*-acting factors regulating translation, together with their associated targets and age-related phenotypes.**

| Molecule | | Regulation upon aging | Targets | Tissue | Age-associated phenotype | Species | References |
|---|---|---|---|---|---|---|---|
| miRNA | *mir-34* | Up | *Eip74EF, Su(z)12, Pcl* | Brain | Vacuolarization, polyglutamine aggregation, and neurodegeneration | Drosophila | Kennerdell et al, 2018; Liu et al, 2012 |
| | | Up | Atg9A | NA | Decreased autophagic flux | Nematode HEK293 Rat | Yang et al, 2013 |
| | | Up | PNUTS | Heart | Increased cardiomyocyte cell death | Mouse | Boon et al, 2013 |
| | *mir-188-3p* | Up | ITGβ3 | endothelial cells | Decreased bone-type H capillary vessel number | Mouse | He et al, 2022 |
| | mir-29 | Up | IRP2 | Brain | Buffered intracellular iron content and oxidative stress | Killifish | Ripa et al, 2017 |
| RBP | Pumilio2 | Down | Mff | Muscle | Altered mitochondria homeostasis | Nematode Mouse | D'Amico et al, 2019 |
| | CPEB1 | Down | Ccnb1 | Oocyte | Decreased oocytes function | Mouse | Takahashi et al, 2023 |
| RNP | Me31B/DDX-6, Imp/ IGF2BP | Increased condensation | *profilin* | Brain | NA | Drosophila | Pushpalatha et al, 2022 |
| | C1q | Increased incorporation in neuronal RNP | 28 targets | Brain | NA | Mouse | Scott-Hewitt et al, 2024 |

*NA* not available.

et al, 2010; Wood et al, 2015); reviewed in (Kinser and Pincus, 2020; Smith-Vikos and Slack, 2012). In these studies, algorithms predicting miRNA targets were used and combined with expression-profiling data to identify numerous transcripts potentially regulated by miRNAs during aging.

As computational-based predictions can generate false positives, it is essential to validate both the miRNA-dependent regulation of potential targets and its functional relevance. So far, this has been done for a few miRNA-target mRNA combinations in both invertebrate and vertebrate model organisms (Kinser and Pincus, 2020). *mir-34*, for example, was shown in *Drosophila* to undergo age-dependent up-regulation in the brain (Liu et al, 2012), inducing an opposite downregulation of its targets Eip74EF (a component of the steroid hormone signaling pathway) and Su(z)12 and Pcl (two components of the PRC2 polycomb repressive complex) (Kennerdell et al, 2018; Liu et al, 2012). Remarkably, *mir-34* mutants exhibit accelerated brain aging characterized on the one hand by early onset, Eip74EF-dependent, vacuolarization and on the other hand by a PRC2-dependent premature sensitivity to polyglutamine aggregation and neurodegeneration (Liu et al, 2012).

As another example, *mir-188-3p* was shown to be upregulated in aged mouse endothelial cells and to negatively regulate the formation of bone-type H capillary vessels (He et al, 2022). While the decline of H-vessel number got alleviated in aged *mir-188-3p* knock-out mice, over-expressing *mir-188-3p* in endothelial cells of young mice decreased H vessels, a process mimicked in vitro by downregulation of the *mir-188-3p* mRNA target ITGβ3 (He et al, 2022).

Together, these and other functional studies have pointed to strong tissue-specificities of miRNAs undergoing age-dependent changes in expression, their target mRNAs and their physiological impacts (Cellerino and Ori, 2017; Kinser and Pincus, 2020; Smith-Vikos and Slack, 2012). Few miRNAs were, however, found to exhibit consistent and evolutionary-conserved regulation during aging. This includes *mir-34*, the expression of which not only increases with age in *Drosophila*, but also in *C. elegans* (de Lencastre et al, 2010), mammalian heart (Boon et al, 2013) and primate brain (Somel et al, 2010). This also includes *mir-29*, which undergoes age-dependent up-regulation in the nervous system of different vertebrates (Baumgart et al, 2012; Ripa et al, 2017; Somel et al, 2010; Takahashi et al, 2012), and was shown in the killifish to downregulate the expression of its target IRP2, thereby limiting intracellular iron concentration upon aging (Ripa et al, 2017).

Although clear examples of miRNAs with conserved functions in regulating the expression of target genes upon aging have been identified, the extent to which miRNAs explain age-dependent RNA-protein level uncoupling remains to be clarified. In *C. elegans*, 30% of the proteins, the levels of which increase with aging, were shown to be sensitive to the loss of function of Dicer, an enzyme required for miRNA processing (Walther et al, 2015). Bioinformatics analyses focusing on gene classes showing discordant RNA-protein profiles in primate brains however revealed that miRNA-binding sites are overall less predictive than RBP-binding sites to separate discordant from concordant classes (Wei et al, 2015). Furthermore, integration of RNA-seq, microRNA-seq and Ribo-seq data from aging killifish brains estimated that changes in miRNA-mediated repression explain less than 7% of the changes in protein abundance over time (Kelmer Sacramento et al, 2020). Although miRNAs each can, in theory, target up to hundreds of mRNAs, they may, in the end, be involved only in specific, yet physiologically

important, regulatory processes during aging. In the future, it will be important to better understand how that specificity is achieved and how aging influences the expression of miRNAs, as our knowledge on the upstream regulatory factors is so far limited.

## Age-dependent changes in levels of RNA-binding proteins

RBPs play a fundamental role in the post-transcriptional regulation of RNA fate. Through binding of selective sets of mRNAs and either masking or recruitment of cellular machineries involved in RNA degradation, translation and/or transport, they regulate the spatio-temporal expression of their target mRNAs (Glisovic et al, 2008). In their bioinformatics analysis of RNA-protein level uncoupling during primate brain aging, Wei et al uncovered a specific and significant enrichment in RBP-binding site number and density—as defined by CLiP experiments—in gene classes showing discordant RNA-protein expression profiles with aging when compared to concordant classes (Wei et al, 2015). Remarkably, different gene classes were enriched for different RBP-binding sites, but gene-class expression patterns during aging were consistent both with the expression profiles of the corresponding RBPs and with the known functions of these RBPs. For example, 85% of the transcripts of one of the defined discordant gene classes, in which protein levels but not RNA levels decrease with aging, contained one or more binding sites for the TIAL1 RBP. Moreover, TIAL1 expression was shown to negatively correlate with target protein levels, consistent with its described function in translational repression (Wei et al, 2015).

Both gene-specific and transcriptome-wide studies have further identified regulatory RBPs whose expression levels change with aging (Chaturvedi et al, 2015; D'Amico et al, 2019; Winsky-Sommerer et al, 2023). In a longitudinal study combining polysome profiling and RNA-seq of aging mouse hippocampus samples, Winsky-Sommerer et al, for example, identified a cluster enriched in RBPs and characterized by a strong past mid-age decrease in expression (Winsky-Sommerer et al, 2023). Upon integration of their multi-omics analyses performed on aging rat brain samples, Ori et al further identified functional protein-interaction networks affected by brain aging and highlighted several networks composed of RBPs involved in RNA degradation, translation or splicing (Ori et al, 2015).

Despite emerging transcriptome-wide sequencing studies predicting the importance of RBP-mediated processes in aging, very few studies have so far functionally explored the role of specific RBPs in driving age-dependent changes in gene expression. The role of Pumilio2 (Pum2), a translational repressor the levels of which were found to be consistently upregulated in various muscle and brain samples from aging mice and *C. elegans* (D'Amico et al, 2019), has however been dissected. In this study, cross-comparison of multi-omics experiments identified the mRNA encoding the mitochondrial fission factor MFF as a candidate Pum2 target. *Mff*, indeed, was found to be bound by Pum2 through a 3'UTR-located Pumilio-binding element (PBE) conserved from mammals to nematodes and to exhibit age-dependent downregulation at the protein but not RNA level. Remarkably, both RNAi-induced depletion of Pum2 ortholog in nematode and Cas9-induced silencing of Pum2 in old mouse muscles led to increased accumulation of the MFF protein and improvement in age-

related alteration of mitochondrial homeostasis, providing strong in vivo evidence of the importance of Pum2-mediated translational repression during aging (D'Amico et al, 2019).

CPEB1 is another RBP whose age-related decline in level was shown in mouse to be causally involved in the altered translation activation of mRNAs essential for the developmental competence of meiosis II-arrested oocytes and fertility (Takahashi et al, 2023). Prematurely reducing the levels of CPEB1 in young oocytes decreased the translation of CPE (cytoplasmic polyadenylation element)-containing mRNAs normally activated during egg maturation, and led to infertility. Conversely, increasing the levels of CPEB1 in aged early-stage oocytes restored both the altered translation pattern of the CPEB1 mRNA target *Ccnb1* and the timing of oocyte maturation (Takahashi et al, 2023). Together, these examples showcase the contribution of RBPs to normal physiological aging. More functional studies are however needed to further illustrate their role and identify their relevant targets.

## Increased condensation of RNA molecules in cytoplasmic RNP granules

Extensive recent work has described and studied the recruitment of tightly regulated mRNAs and their associated proteins into large cytoplasmic assemblies termed RNA condensates or RNP (ribonucleoprotein) granules (Ripin and Parker, 2023). Different types of granules, with overlapping yet distinct RNA and protein content have been defined: some are constitutive (e.g., P-bodies, neuronal RNP granules), others are induced by stress (e.g., Stress Granules) (An et al, 2021; Bauer et al, 2023; Ripin and Parker, 2023). Condensation of RNA molecules into cytoplasmic granules has been implicated in RNA buffering, translational repression or RNA compartmentalization (Adekunle and Hubstenberger, 2020; Putnam et al, 2023). Under normal conditions, cytoplasmic RNP granules undergo highly dynamic molecular turnover. In the context of age-related diseases such as neurodegenerative diseases, static pathological inclusions enriched in RNP granule components, including ALS disease-causing RBPs such as TDP-43, FUS or hnRNPA1, have repeatedly been observed (Ling et al, 2013). These observations, combined with in vitro studies showing that mutant RBPs assemble into less dynamic assemblies, led to a model in which abnormal aggregation of granule components may occur over time, favored by disease mutations and the local concentration of interaction-prone molecules (Alberti and Hyman, 2021; Bauer et al, 2023; Kiebler and Bauer, 2024). As these studies were performed in disease contexts, it had remained unclear until recently if and how aging impacts on RNP granules in healthy contexts.

Proteomic analysis of the insoluble proteome has however revealed a propensity of RBPs, and particularly RBPs with disordered prion-like domains such as the helicase DDX-5, to aggregate in the aging killifish brains (Harel et al, 2024). Similarly, work performed in gonad-less aging *C.elegans* has identified proteins whose insolubility increase with age in control individuals, but not in long-lived *daf-2* individuals with reduced insulin signaling, highlighting a high over-representation of RNP granule components (Lechler et al, 2017). Age-dependent decrease in RNP granule component solubility was however not systematically observed (Kelmer Sacramento et al, 2020; Molzahn et al, 2023), which may reflect differences of biochemical purifications, time

windows and/or tissue-specificity. Indeed, systematic comparison of aggregation patterns across various killifish organs revealed very little overlap in the aggregating proteomes isolated from brain, gut, liver, heart, muscle, skin, and testis (Chen et al, 2024).

At the cellular level, different studies have described an increased recruitment of RBPs into cytoplasmic RNP granules with aging. In yeast, the number of cells with visible P-bodies was shown to increase in the course of replicative aging (Hu et al, 2018; Rieckher et al, 2018). In mouse neocortex tissues, an increased accumulation of Pum2 into cytoplasmic granules was observed in aged individuals (D'Amico et al, 2019). In *C. elegans*, the Stress Granule components PAB-1 and TIAR-2, which are diffusely localizing in pharyngial muscles of young animals, were found to accumulate in Stress Granule-like solid aggregates from mid-age (7 days) onwards, with nearly half of the end-stage individuals showing significant aggregate accumulation (Lechler et al, 2017). As revealed by a detailed analysis performed in *Drosophila* brain, the accumulation of Stress Granule-like punctae may represent an end-of-live phenomenon, as no such accumulation was observed in mid-aged individuals (De Graeve et al, 2022; Pushpalatha et al, 2022). Rather, conserved components of constitutive RNP granules (e.g., Me31B/DDX-6 and Imp/IGF2BP) were found to gradually accumulate into larger, yet dynamic, granules distinct from solid aggregates with aging (Pushpalatha et al, 2022). Age-related condensation was induced by increased levels of the granule nucleator Me31B, and required the activity of the PKA signaling pathway. Furthermore, RBP-bound mRNAs also exhibited increased, 3'UTR-mediated, condensation into RNP granules, which triggers their translational repression with aging.

These results suggest that the recruitment of selected mRNAs to repressive granules of increased size and number may represent a mechanism underlying transcript-specific post-transcriptional regulation in the context of aging. Consistent with a regulatory function of RNA condensation, recent work discovered that the innate immune complement protein C1q is upregulated in microglial cells of the aged mouse brain and internalized by neuronal cells to integrate into neuronal RNP granules rich in ribosomes and to regulate the translation of specific sets of proteins (Scott-Hewitt et al, 2024). This function appears to be independent of the complement pathway, but how it is mediated remains to be investigated.

### Increased translation pausing at specific positions

The above-mentioned examples point to the importance of regulatory elements in the non-translated regulatory 5' and 3'UTR regions of mRNAs in driving age-dependent post-transcriptional regulation. Features present in the coding sequences of specific mRNAs were also recently shown to alter the kinetics of translation elongation and the production of associated proteins. Stein et al, uncovered through Ribo-seq analyses in yeast and *C. elegans* a specific age-related increase in ribosome pausing at positions coding for Proline as well as basic Arginine and Lysine residues (Stein et al, 2022). Pausing was particularly exacerbated at polybasic stretches, where it associated with both increased ribosome collision and increased aggregation of the elongating peptide chain, a process triggered by a deficient ribosome-associated quality-control (RQC) pathway likely overwhelmed in the context of aged cells. Remarkably, a similar conclusion was reached in a multi-omics study performed in killifish brains that observed increased ribosome pausing and collision at codons encoding

basic residues (Di Fraia et al, 2024). In this study, modeling also suggested that elongation pausing may be an important process driving the uncoupling of RNA and protein levels, particularly affecting a proteome subset enriched in basic residues. It will now be interesting to investigate why ribosome pausing increases specifically at these positions in aging organisms and tissues.

## Concluding remarks

The various physiological changes characterizing the aging process are accompanied by alterations in both tissue-specific and systemic gene expression programs. Although most of the observed changes in gene expression correspond to small variations in absolute level, a significant fraction of the genome undergoes strong age-dependent variations in expression levels that cannot simply be explained by a decreased efficiency of the different cellular machineries involved in gene expression (Cellerino and Ori, 2017).

Control of RNA translation has emerged as an important post-transcriptional mechanism explaining both global and gene-specific expression changes with age. Molecular players contributing to translation regulation in the context of aging have been identified, notably translation initiation and elongation factors, miRNAs or RNA-binding proteins. The role of other molecular pathways, the activity of which varies with age, now remains to be investigated. m6A RNA methylation patterns were shown in different tissues and organisms to be modified with aging (Castro-Hernandez et al, 2023; Jiang et al, 2021; Perlegos et al, 2024; Wu et al, 2023a; Wu et al, 2023b) which suggests their contribution to changes in RNA stability and translation (Perlegos et al, 2024; Tassinari et al, 2023). More recent work has however challenged the view that m6A marks influence RNA translation (Zaccara and Jaffrey, 2024), thus questioning the importance of this process in the control of age-dependent changes in translation profiles. Long noncoding RNAs have for years been identified as important players of aging (Grammatikakis et al, 2014), but have only recently been suggested to regulate this process through translation regulation (Anver et al, 2024), opening new perspectives related to their contribution to age-dependent translation changes. Lastly, alternative splicing site selection has also been shown to vary with age in various tissues and organisms (Bhadra et al, 2020; Ham et al, 2022; Kumar et al, 2024; Mazin et al, 2013; Rodriguez et al, 2016; Tollervey et al, 2011; Ubaida-Mohien et al, 2019), causing isoforms with different UTR length, *cis*-regulatory elements and post-transcriptional dynamics (Kumar et al, 2024). How such changes functionally contribute to age-dependent translational regulation remains to be investigated.

More generally, a number of challenges remain to be addressed to better characterize the post-transcriptional mechanisms driving changes in gene expression relevant to aging physiology (Box 2). First, one needs to go beyond correlative analyses and establish functional causality links. This can be achieved using modeling approaches (Di Fraia et al, 2024; Janssens et al, 2015), but requires transcriptome-wide data sets of high temporal resolution and replicate numbers, as increased variability in RNA and protein levels is observed in aging tissues and organisms (Cellerino and Ori, 2017). This can also be achieved through functional studies which inactivate *trans*-acting machineries or *cis*-regulatory elements, and assess the direct impact on gene expression in vertebrate and/or invertebrate model organisms (Box 1).

**Box 2  In need of answers**

  i. Are there sex-specific differences in age-related post-transcriptional regulatory processes?
 ii. Is the age-dependent uncoupling between protein and RNA levels progressive or are there critical periods?
iii. What are the key RNA-binding proteins mediating post-transcriptional changes in gene expression upon aging? Do they control conserved "regulons" contributing to physiological aging?
 iv. Are the changes in gene expression regulated post-transcriptionally detrimental or protective in the context of aging? Are these changes involved in translational buffering?
  v. Does the age-dependent condensation of RNAs and associated proteins represent a seed for the assembly of pathological inclusions observed in late-onset degenerative diseases? More generally, how independent are the age-related changes in post-transcriptional regulatory mechanisms from the disease-causing mechanisms?

Second, one needs to understand if the observed post-transcriptional changes are protective or if they rather participate to the decline in biological functions occurring with age. Although both scenarios are likely to be at play in a tissue-, organism- and gene-specific manner, it is interesting to consider that post-transcriptional regulation, and in particular translation regulation, may have buffering functions (Kusnadi et al, 2022). Translational compensation mechanisms have indeed been shown in different species to maintain proteome composition in contexts with initial differences in mRNA abundance (Cenik et al, 2015; Kusnadi et al, 2022; McManus et al, 2014). Lastly, one needs to better understand if and how age-dependent post-transcriptional changes contribute to disease susceptibility. Consistent with shared common mechanisms, genetic mutations in translation regulatory factors and ribosomal components were causally implicated in a number of age-related diseases (Tahmasebi et al, 2018), including cancer (Robichaud et al, 2019) and neurodegenerative diseases (Skariah and Todd, 2021).

# Peer review information

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

## Acknowledgements

The authors apologize to those whose work could not be cited owing to space constraints. This study was supported by the CNRS, as well as grants from the INCA (2021-162), the ANR (ANR22-CE12-0024), and the CEFIPRA (IFC/6503-E/2021/193). MS received fellowships from the French Ministry of Research and the ARC Foundation (ARCDOC42024010007708). The authors thank Hiba Laghrissi for the critical reading of the manuscript.

## Author contributions

**Mathilde Solyga**: Writing—original draft; Writing—review and editing.
**Amitabha Majumdar**: Writing—original draft; Writing—review and editing.
**Florence Besse**: Conceptualization; Funding acquisition; Writing—original draft; Project administration; Writing—review and editing.

## Disclosure and competing interests statement

The authors declare no competing interests.

