## [Peer Review File · EMBO Reports]

Regulating translation in aging: from global to gene-specific mechanisms

Mathilde Solyga, Amitabha Majumdar, and Florence Besse

Corresponding author(s): Florence Besse (florence.besse@univ-cotedazur.fr)

Review Timeline:

Submission Date:	29th Aug 24
Editorial Decision:	8th Oct 24
Revision Received:	23rd Oct 24
Accepted:	29th Oct 24

Editor: *Esther Schnapp / Holger Breithaupt*

Transaction Report:

Dear Florence,

Thank you for the submission of your Review to EMBO reports. We have finally received all 3 referee reports, and I have never seen so short reports for any review. This is probably a good sign :-)

All referees rate the interest, novelty and clarity of this review "high", which is great. Only referees 1 and 2 have some suggestions for how the review could be improved, and I think all suggestions are good and should be addressed. Please let me know in case you disagree.

As for timing, if you could submit the revised review before the 4th of November and if you provide high quality figures that do not need to be redrawn by our graphics designer, we could still publish your review in our December issue, which would be my preferred option. The deadline for our December issue is the 7th of November, and I need at least one full day for the editing of the review and you will probably need a day for approving my edits.

Otherwise, which is also fine, you can submit the revised review in early November and I will send the figures to our graphics designer who will need 10 full working days to redraw the figures. We can then publish your review in our January issue. Please let me know what you prefer.

Thank you again for contributing this nice piece to EMBO reports !

With best wishes,
Esther

Referee #1:

This is a timely review of changes in translation with age, which may explain the uncoupling between RNA and protein levels during ageing. I think this is an interesting review, which is well-written and I have no major concerns.

One minor issue is that at times it is not clear which species the authors are referring to in their statements, for example on page 7: "First, levels of ribosomal proteins were found to be strongly downregulated in aging tissues..." - I suggest making this clearer throughout.

Page 20, for a paper on miRNA profiling of aging in rat brains:
<https://pubmed.ncbi.nlm.nih.gov/26694192/>

Overall, I think this is an interesting and timely work that will make a fine contribution to the literature.

It is my usual policy to reveal my identity to the authors: Joao Pedro de Magalhaes.

Referee #2:

The review by Solyg and colleagues cover an important yet not fully appreciated mechanism of aging, which is the regulation of protein synthesis. The topic is timely and the review is well written and comprehensive. I have only two minor comments:

- The Figures should be improved. Especially Figure 2 and 3 are hard to interpret. A more detailed graphical representation of the mechanisms contributing to regulation of protein translation should be provided and their consequences. It would be additionally beneficial to complement the figures with a summary table of the aging phenotypes related to translation and the model organism and cell types in which these have been described.
- In paragraph 3.2, it should be mentioned that mutations in RBPs are also linked to age-related diseases, e.g., TDP-43 in neurodegeneration.

Referee #3:

In this review, Solyga et al. provide an overview of the role of mRNA translation alterations in the aging processes. The authors

describe the progressive decoupling of RNA and protein levels observed during aging in various biological systems and then address different mechanisms that could explain this phenomenon, broadly divided into two categories: global and transcript-specific. The paper is timely, clear, and meticulously analyzes the state of the art. It will be of great interest to both the aging and RNA research communities. I am pleased to recommend its publication in EMBO Reports.

Point by point answer to the referees

Below you will find our detailed answers to the points raised by the referees.

Please note that the modifications we have introduced in the text are visible with the correction mode.

Referee #1 -----

1) One minor issue is that at times it is not clear which species the authors are referring to in their statements, for example on page 7: "First, levels of ribosomal proteins were found to be strongly downregulated in aging tissues..." - I suggest making this clearer throughout.

We carefully checked throughout the manuscript and identified 4 instances in which species were not explicitly mentioned (page 5, 6, 7 and 8). This has been corrected.

2) Page 20, for a paper on miRNA profiling of aging in rat brains:
<https://pubmed.ncbi.nlm.nih.gov/26694192/>

This reference (Wood et al., 2015) has been added in page 10, at the beginning of the section entitled "miRNA in aging".

Referee #2 -----

I have only two minor comments:

- The Figures should be improved. Especially Figure 2 and 3 are hard to interpret. A more detailed graphical representation of the mechanisms contributing to regulation of protein translation should be provided and their consequences.

We have revised Figures 2 and Figure 3, so as to better describe the molecular processes altered upon aging and their molecular outputs.

In Figure 2, we have added 2 levels of text below the schematic representation of a translating mRNA: the first one describes the process impacted upon aging (e.g. "decreased initiation") and the second one, in bold describes the resulting output (e.g. "less ribosomes per transcript").

In Figure 3, we have better separated and annotated the 3 molecular contributors to gene-specific translational repression upon aging ("miRNA", "RBP", "RNP granules"). We have also simplified our representation of each process and explicitly indicated on the Figure changes seen upon aging (e.g. "increased miRNA level"). Last, we have more clearly indicated at the bottom of the panel the translation status (active translation in young, decreased translation in aged condition).

Please note that we have also modified the corresponding Figure legends to help the reader better understand the processes displayed.

It would be additionally beneficial to complement the figures with a summary table of the aging phenotypes related to translation and the model organism and cell types in which these have been described.

We are now providing a summary Table that lists the studies in which an age-dependent molecular change has been associated with a gene-specific alteration of translation and a subsequent phenotypic change. This table is referred to at the end of page 9.

- In paragraph 3.2, it should be mentioned that mutations in RBPs are also linked to age-related diseases, e.g., TDP-43 in neurodegeneration.

In the revised version, we now mention 3 RBPs whose mutations have been linked to age-related neurodegenerative diseases: TDP-43, FUS and hnRNPA1. These proteins are not mentioned in paragraph 3.2, which is dedicated to RBPs whose endogenous levels change upon healthy aging, but rather in paragraph 3.3, where the presence of these 3 proteins in the pathological aggregates we described is now mentioned (page 13).

Referee #3 -----

Referee #3 had no point.

Dr. Florence Besse
Institut de Biologie Valrose
CNRS-UMR 7277
Centre de Biochimie
Parc Valrose
Nice, Cedex 2 06108
France

Dear Dr. Besse,

I am pleased to inform you that your manuscript has been accepted for publication in EMBO reports. Your manuscript will be processed for publication by EMBO Press. It will be copy edited and you will receive page proofs prior to publication. Please note that you will be contacted by Springer Nature Author Services to complete licensing information.

Yours sincerely,

Holger Breithaupt, PhD
Senior Editor, Science & Society
EMBO reports
